# Heterogeneity-Aware Federated Deep Multi-View Clustering towards Diverse Feature Representations

Xiaorui Jiang
CCCD Key Lab of Ministry of Culture
and Tourism, University of Science
and Technology of China
Hefei, China
xrjiang@mail.ustc.edu.cn

Zhongyi Ma
CCCD Key Lab of Ministry of Culture
and Tourism, University of Science
and Technology of China
Hefei, China
mzymail@mail.ustc.edu.cn

Yulin Fu
CCCD Key Lab of Ministry of Culture
and Tourism, University of Science
and Technology of China
Hefei, China
fuyulin@mail.ustc.edu.cn

Yong Liao*
CCCD Key Lab of Ministry of Culture
and Tourism, University of Science
and Technology of China
Hefei, China
yliao@ustc.edu.cn

Peng Yuan Zhou*
Department of Electrical and
Computer Engineering, Aarhus
University
Aarhus, Denmark
pengyuan.zhou@ece.au.dk

## Abstract

Multi-view clustering has proven to be highly effective in exploring consistency information across multiple views/modalities when dealing with large-scale unlabeled data. However, in the real world, multi-view data is often distributed across multiple entities, and due to privacy concerns, federated multi-view clustering solutions have emerged. Existing federated multi-view clustering algorithms often result in misalignment in feature representations among clients, difficulty in integrating information across multiple views, and poor performance in heterogeneous scenarios. To address these challenges, we propose HFMVC, a heterogeneity-aware federated deep multi-view clustering method. Specifically, HFMVC adaptively perceives the degree of heterogeneity in the environment and employs contrastive learning to explore consistency and complementarity information across clients' multi-view data. Besides, we seek consensus among clients where local data originates from the same view, incorporating a contrastive loss between local models and the global model during local training to adjust consistency among local models. Furthermore, we elucidate the sample representation logic for local clustering in different heterogeneous environments, identifying the degree of heterogeneity by computing the within-cluster sum of squares (WCSS) and the average inter-cluster distance (AICD). Extensive experiments verify the superior performance of HFMVC across both IID and Non-IID settings.

*Corresponding authors.

## CCS Concepts

• **Computing methodologies** → **Cluster analysis**; • **Theory of computation** → **Unsupervised learning and clustering**.

## Keywords

Multi-view clustering; federated learning; contrastive learning

**ACM Reference Format:**
Xiaorui Jiang, Zhongyi Ma, Yulin Fu, Yong Liao, and Peng Yuan Zhou. 2024. Heterogeneity-Aware Federated Deep Multi-View Clustering towards Diverse Feature Representations. In *Proceedings of the 32nd ACM International Conference on Multimedia (MM '24), October 28–November 1, 2024, Melbourne, VIC, Australia.* ACM, New York, NY, USA, 10 pages. https://doi.org/10.1145/3664647.3681302

## 1 Introduction

Perceiving the same thing from multiple perspectives often allows for a more comprehensive understanding of the information. In recent years, the rapid development of multimedia technology [10, 12] has generated a substantial amount of unlabeled multimodal data, posing a challenge for clustering analysis of multi-view data. In response, Multi-View Clustering (MVC) [2, 3, 5, 6, 11] methods have been proposed and gained momentum. These methods expedite the exploration of consistent and complementary information within the data by fostering collaboration across multiple views, thereby achieving a comprehensive clustering structure.

However, most existing MVC methods [23, 41, 44, 46] are built on the assumption of centralization, presuming that multi-view data is stored within a single entity, which proves challenging to apply in many real-world scenarios. For example, patients may undergo relevant examinations at different hospitals. Only through the aggregation of the medical data can their health status be accurately reflected. However, the simplistic consolidation of this data may give rise to privacy concerns [25]. To address this challenge, Federated Learning (FL) [17, 35] presents a promising approach by aggregating individual client models through a central server while preserving the privacy of raw data. The integration of FL with MVC, referred to as Federated Multi-View Clustering (FedMVC)

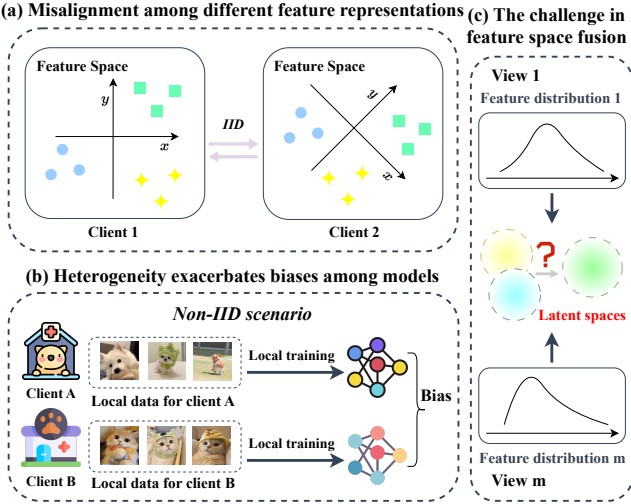

**Figure 1: Problem illustration of existing FedMVC methods.**

[8], which aims to explore a more comprehensive clustering structure from unlabeled multi-view data distributed across multiple clients while preserving privacy [21]. Despite the demonstrated potential of FedMVC in distributed environments, the amalgamation of these technologies has not garnered sufficient attention due to three crucial challenges, as depicted in Figure 1.

(1) ***Challenge 1***: Due to the heterogeneity of local data, randomness in the training process, and differences in training time and computational resources among clients, there are inconsistencies in the feature representations encoded by the clients. Specifically, data representations on each client may exhibit angular deviation [51], so an effective method is needed to align these representations.

(2) ***Challenge 2***: Heterogeneity intensifies this misalignment, as data from different clients may vary in terms of categories and quantities. While some methods [26, 28, 30, 53] address the challenge of non-independent and identically distributed (Non-IID) data in federated learning, they fall short when it comes to handling unsupervised learning tasks, making them difficult to apply directly.

(3) ***Challenge 3***: Multi-view data may be distributed across multiple clients, resulting in non-uniformity between the feature spaces encoded by each client. Therefore, efficiently performing multi-view information mining across multiple clients is challenging.

***Our solution***: To address the aforementioned challenges, we propose HFMVC, a Heterogeneity-aware Federated deep Multi-View Clustering method. HFMVC is designed for distributed scenarios where each client possesses data from a single view. Specifically, each client learns representations specific to a single view by deploying an autoencoder to encode and reconstruct its local data. Then we map the encoded features from all views into a unified high-level space to eliminate irrelevant information. Based on this, we address the aforementioned challenges through the following methods: (1) To tackle the misalignment between representations generated by different clients mentioned in ***Challenge 1***, we employ selective aggregation, wherein the server aggregates only the local models uploaded by clients with the same view data at a given time. This allows knowledge to be shared among these clients, improving

their ability to represent data from the same view and aligning their encoded features in the feature space. (2) To mitigate the negative impact of heterogeneous data on feature representation mentioned in ***Challenge 2***, our objective is to reveal the differences in feature representations across different heterogeneous environments and adopt targeted feature learning approaches. By evaluating the within-cluster sum of squares ($WCSS$) and the average inter-cluster distance ($AICD$) of the concatenated high-level features, we can measure the degree of heterogeneity and assess the training trends of local models, thus conducting model-level contrastive learning. (3) To address the challenge of conducting feature learning across multiple clients mentioned in ***Challenge 3***, we use a trusted server to facilitate the exchange of high-level features. The dual contrastive learning module allows each client to learn multi-view features, ultimately leading to the exploration of clearer clustering structures. Through these measures, we can evaluate the heterogeneity of the environment based on the characteristics of the initial local clustering and promote the flow of information among clients to fully utilize their knowledge, ultimately discovering a clear clustering structure. In summary, our contributions primarily include:

- We propose HFMVC, a novel federated MVC method with robust generalization capabilities, enabling the exploration of clear clustering structures from multiple clients.
- We introduce a heterogeneity-aware module that can assess the degree of data heterogeneity based on the local clustering results from the clients. This enables HFMVC to adaptively adjust its clustering strategy, achieving robust adaptability in both IID and Non-IID scenarios.
- We conduct extensive experiments across varying degrees of heterogeneity and different numbers of clients. The results demonstrate that HFMVC consistently exhibits state-of-the-art performance while ensuring privacy.

## 2 Related Work

### 2.1 Federated Multi-View Clustering

As a subdomain within federated multi-view learning [14, 15, 22, 49], federated multi-view clustering is designed to execute unsupervised clustering tasks in multi-view settings by fostering collaboration among numerous clients. Due to the Non-IID data among different clients, the models trained by them exhibit variations, consequently leading to distinct feature representations. It is precisely this departure from traditional centralized MVC methods [29, 32, 39, 41, 48, 50, 52] that makes addressing the challenges of FedMVC highly demanding. Hu et al. [21] proposed FedMVFPC, a federated multi-view fuzzy clustering method to address the challenges of feature heterogeneity and distributed data storage. However, compared to deep MVC, this strategy based on traditional methods inherently lacks robust feature representation capabilities. Moreover, as the number of clients increases, FedMVFPC experiences significant performance degradation. Chen et al. [8] proposed FedDMVC, a federated deep multi-view clustering method designed to tackle the challenge of incomplete multi-view data in distributed environments through global self-supervision. However, FedDMVC assumes precise partitioning of data from $M$ views among $M$ clients, suggesting that the number of clients in this FL system strictly equals the number of views, thereby limiting its

applicability. Furthermore, FedDMVC overlooks the challenge of high communication costs that may lead to server paralysis, further impacting its availability. These identified weaknesses motivate us to propose a more sophisticated and practical MVC solution.

## 2.2 Federated Unsupervised Representation

Unsupervised learning methods can be classified into generative and discriminative types. The former focuses on learning representations by generating pixels in the input space [18, 19, 27], while the latter involves modeling data to discover intrinsic patterns or relationships within it [1, 16, 36]. Some related studies [24, 40, 40] have attempted to apply FL to unlabeled data. However, a straightforward combination of FL with unsupervised methods may not yield satisfactory results. Lu et al. [33] proposed FedUL, a federated unsupervised learning framework. But FedUL requires specific prior knowledge, which often poses challenges in practice. Liao et al. [31] proposed FedU$^2$ to analyze representation in federated unsupervised learning with Non-IID data. Zhang et al. [51] proposed FedCA, a method based on federated contrastive averaging with a dictionary and alignment. While FedCA overlooks privacy concerns and shows limited improvement in experimental results. Besides, there is limited research on the data representation problem in federated multi-view scenarios involving multiple clients.

## 3 Methodology

### 3.1 Problem Definition

Here we present the formal definition of federated multi-view clustering. Suppose there is a dataset with $N$ samples and $M$ views distributed among $C$ clients (denoted as $\mathbf{X} = \{\mathbf{X}^1, \mathbf{X}^2, ..., \mathbf{X}^C\}$), with an expectation to be partitioned into $K$ clusters. Each client possesses a subset of the data corresponding to a particular view $\mathbf{X}^c \in \mathbb{R}^{N_c \times D_m}$, where $N_c$ and $D_m$ represent the number of samples in client $c$ and the dimensionality of samples in view $m$, respectively. The local models with corresponding local data on the client-side are defined as $\mathcal{F} = \{F_1(\mathbf{X}^1; \mathbf{w}^1), F_2(\mathbf{X}^2; \mathbf{w}^2), \cdots, F_C(\mathbf{X}^C; \mathbf{w}^C)\}$. We use widely adopted autoencoders [9, 23, 44, 47, 48] for data representation and reconstruction. Specifically, each local model consists of the most basic encoder-decoder pair $\{f^c_{\phi^c}(\cdot), g^c_{\theta^c}(\cdot)\}$, which denote the encoding and decoding processes for client $c$, where the learnable parameters are represented by $\phi^c$ and $\theta^c$, respectively. Autoencoders can map raw data to a specified feature space, i.e., $f^c_{\phi^c}(\mathbf{X}^c; \phi^c) : \mathbf{X}^c \in \mathbb{R}^{N_c \times D_m} \longmapsto \mathbf{Z}^c \in \mathbb{R}^{N_c \times d_m}$ and $g^c_{\theta^c}(\mathbf{Z}^c; \theta^c) : \mathbf{Z}^c \in \mathbb{R}^{N_c \times d_m} \longmapsto \hat{\mathbf{X}}^c \in \mathbb{R}^{N_c \times D_m}$. Here, $N_c$ is the number of samples for client $c$, and $d_m$ is the dimensionality of the features after encoding. Please note HFMVC assumes each client only has data from a single view, yet it still supports more scenarios by deploying multiple autoencoders on each client (please see the Appendix). Based on this, we introduce two additional definitions:

(1) **Friend clients**: Refers to clients whose samples correspond to the same object but come from different views (e.g., client 1 and client n-1 in Figure 2 are friend clients).

(2) **Peer clients**: Refers to clients whose samples correspond to different objects but come from the same view (e.g., client n-1 and client n in Figure 2 are peer clients).

## 3.2 Representation Learning

We instruct each client to reconstruct their local data to obtain client-specific feature representations. The reconstruction loss is:

$$\mathcal{L}_R = \sum_{c=1}^{C} \mathcal{L}_R^c = \sum_{c=1}^{C} \left\| \mathbf{X}^c - \hat{\mathbf{X}}^c \right\|_2^2 = \sum_{c=1}^{C} \sum_{i=1}^{N_c} \left\| \mathbf{x}_i^c - g^c_{\theta^c}\left(f^c_{\phi^c}(\mathbf{x}_i^c)\right) \right\|_2^2. \tag{1}$$

Due to the potential noise in the encoded features from individual client data and the possibility that data from different clients might be encoded into different latent spaces, we need to further learn high-level features within the same feature space. Therefore, we add an additional MLP layer $\mathcal{P}^{(c)}$ to the autoencoder model of client $c$, aiming to explore more valuable information. Here, we use $\mathcal{H} = \{\mathbf{H}^1, \mathbf{H}^2, ..., \mathbf{H}^M\}$ to represent high-level features. $\{\mathbf{H}_j^m\}_{j=1}^{|C_m|}$ denotes the high-level features of view $m$ distributed across $|C_m|$ clients, where $C_m$ is the set of clients with data from view $m$ and $\mathbf{h}_i^c = \mathcal{P}^{(c)}(\mathbf{z}_i^c)$ represents the high-level features extracted from the encoded feature $\mathbf{z}_i^c$ for sample $i$ of client $c$, i.e., $\sum_{m=1}^{M} |C_m| = C$.

## 3.3 Dual Contrastive Learning

The objective of HFMVC is to achieve more comprehensive information acquisition through collaboration among multiple clients. Based on this, HFMVC performs selective aggregation among **peer clients** and dual contrastive learning among **friend clients**.

*3.3.1 Selective aggregation among peer clients.* Since the data of peer clients only vary in quantity and category, the server can selectively aggregate their local models to enhance each client's feature representation capabilities, thereby mitigating the negative impact of heterogeneous data. Thus the global model for view $m$ is:

$$\bar{\mathbf{w}}_m = \frac{1}{|C_m|} \sum_{i \in C_m} \mathbf{w}^i. \tag{2}$$

*3.3.2 Contrastive learning among friend clients.* Given that data among friend clients originates from different views but describes the same object, extracting complementary information between them proves advantageous for uncovering crucial semantics. Inspired by [45], we implement contrastive learning among friend clients. Specifically, the server forwards the high-level features uploaded by client $c$ to its friend client(s). Following this procedure, each client acquires the high-level features of its friend client(s). Then each high-level feature $\mathbf{h}_i^c$ from client $c$ generates $(MN_c - 1)$ feature pairs, denoted as $\{\mathbf{h}_i^c, \mathbf{h}_j^p\}_{j=1,...,N_c}^{p=1,...,|C'|}$, where $C'$ represents the set of friends for this client, satisfying $|C'| = M$. And these pairs include $(M - 1)$ positive pairs $\{\mathbf{h}_i^c, \mathbf{h}_i^p\}_{c \neq p}$ and the remaining $M(N_c - 1)$ negative pairs. Besides, cosine similarity is commonly employed as a measure of similarity for high-level features [45, 46]:

$$\text{sim}\langle \mathbf{h}_i^c, \mathbf{h}_j^p \rangle = \frac{\mathbf{h}_i^c \cdot \mathbf{h}_j^p}{\|\mathbf{h}_i^c\| \|\mathbf{h}_j^p\|}. \tag{3}$$

We use $\tau_C$ to denote a temperature parameter, thus the overall contrastive loss for the clients possessing data from view $c$ and those possessing data from view $p$ is:

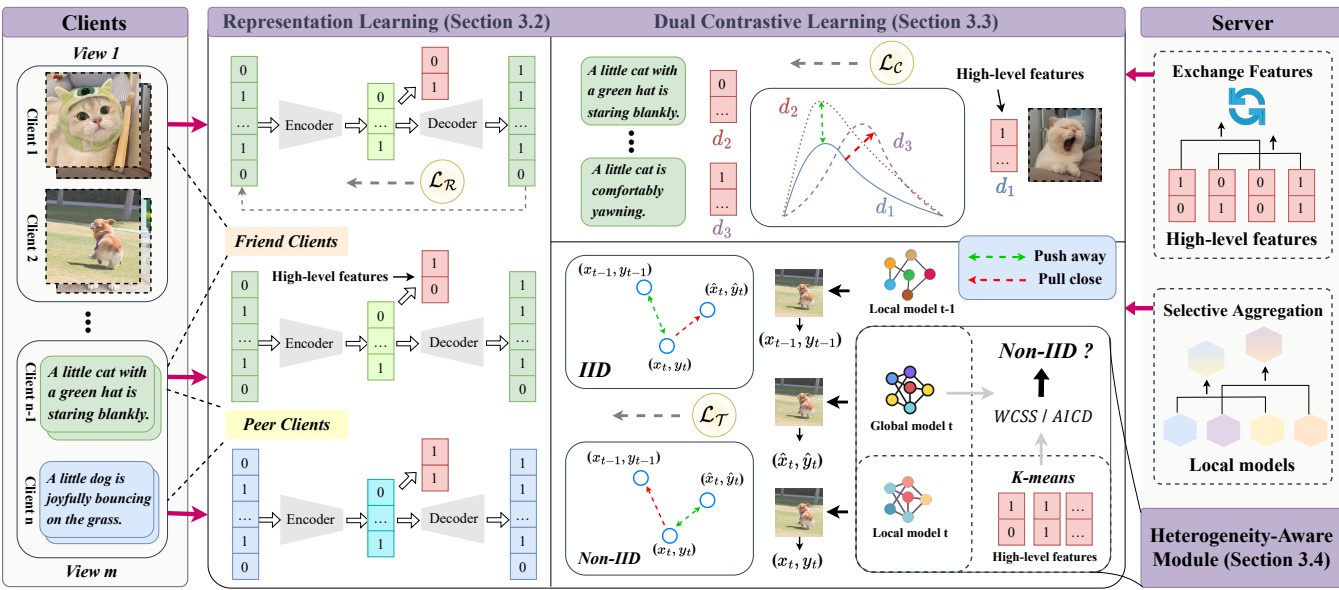

**Figure 2: The framework of HFMVC. After the representation of samples from each client (Section 3.2), the system assesses the heterogeneity in the environment (Section 3.4), and adopts dual contrastive learning (Section 3.3).**

$$\mathcal{L}_C^{(cp)} = -\frac{1}{N} \sum_{i=1}^{N} \log \frac{e^{\mathbf{sim}\langle \mathbf{h}_i^c, \mathbf{h}_i^p \rangle / \tau_C}}{\sum_{j=1}^{N} \sum_{v=c,p} e^{\mathbf{sim}\langle \mathbf{h}_i^c, \mathbf{h}_j^v \rangle / \tau_C} - e^{1/\tau_C}}. \quad (4)$$

Since all clients perform the above steps, from a global perspective, the total contrastive loss can be accumulated as the sum of contrastive losses for all clients, and it can be represented as:

$$\mathcal{L}_C = \frac{1}{2} \sum_{m=1}^{M} \sum_{n \neq m} \mathcal{L}_C^{(mn)}. \quad (5)$$

By conducting contrastive learning among friend clients, HFMVC achieves the exploration of common semantics across views.

*3.3.3 Contrastive learning in local training.* The key challenge posed by the distributed environment to MVC is the lack of uniformity in representation spaces across different clients. Although HFMVC addresses this by selectively aggregating among peer clients to uniformly represent the feature space, biases may still emerge after multiple rounds of local training. To tackle this issue, a promising strategy involves integrating contrastive learning into the local training process on the client side [7, 28].

Specifically, in the $t$-th round of global training, client $c$ is required to independently extract representations for local data $\mathbf{X}^c$ using the selectively aggregated view-specific encoder $f_\phi^{g,t}(\cdot)$, local encoder $f_{\phi^c}^{c,t}(\cdot)$ in round $t$ and local encoder $f_{\phi^c}^{c,t-1}(\cdot)$ in round $t-1$. Subsequently, client $c$ can obtain the corresponding high-level feature representations $\mathbf{H}^{g,t} = \mathcal{P}^{(g,t)}\left(f_\phi^{g,t}(\mathbf{X}^c;\phi)\right)$, $\mathbf{H}^{c,t} = \mathcal{P}^{(c,t)}\left(f_{\phi^c}^{c,t}(\mathbf{X}^c;\phi^c)\right)$ and $\mathbf{H}^{c,t-1} = \mathcal{P}^{(c,t-1)}\left(f_{\phi^c}^{c,t-1}(\mathbf{X}^c;\phi^c)\right)$. Then, we adopt different client-level contrastive learning strategies based on the system's degree of heterogeneity.

(1) In IID scenarios, the sample distributions among peer clients are similar. However, due to factors such as randomness during training, different models may generate representations with certain angular deviations, resulting in misalignments between representations. Nevertheless, the model obtained through selective aggregation integrates knowledge from multiple clients, leading to a model with higher representational capacity. Therefore, we propose defining the model-contrastive loss:

$$\mathcal{L}_T^{(c,iid)} = -\log \frac{e^{\mathbf{sim}\langle \mathbf{H}^{c,t}, \mathbf{H}^{g,t} \rangle / \tau_T}}{e^{\mathbf{sim}\langle \mathbf{H}^{c,t}, \mathbf{H}^{g,t} \rangle / \tau_T} + e^{\mathbf{sim}\langle \mathbf{H}^{c,t}, \mathbf{H}^{c,t-1} \rangle / \tau_T}}. \quad (6)$$

Here, $\tau_T$ represents the temperature parameter. The significance of Eq. (6) is to treat $(\mathbf{H}^{c,t}, \mathbf{H}^{g,t})$ as a positive pair and $(\mathbf{H}^{c,t}, \mathbf{H}^{c,t-1})$ as a negative pair. This allows the model to bring local representations closer to global representations, and amplify the changes in feature representations before and after each round of local training.

(2) In Non-IID scenarios, since each client has different quantities and categories of local samples, the representation capabilities of clients for each class of samples vary. The effectiveness of a client's representation of a certain data class depends on the quantity of samples it possesses for that class. Consequently, all clients exhibit strong, and sometimes unique, representation capabilities for the majority class in their local data. Therefore, during local training, it is crucial for the client's feature representation to maintain a greater distance from the representation generated by the global model. Additionally, minimizing the change in feature representation before and after each round of local training is necessary. Thus, the model-contrastive loss is reformulated as:

$$\mathcal{L}_T^{(c,noniid)} = -\log \frac{e^{\mathbf{sim}\langle \mathbf{H}^{c,t}, \mathbf{H}^{c,t-1} \rangle / \tau_T}}{e^{\mathbf{sim}\langle \mathbf{H}^{c,t}, \mathbf{H}^{g,t} \rangle / \tau_T} + e^{\mathbf{sim}\langle \mathbf{H}^{c,t}, \mathbf{H}^{c,t-1} \rangle / \tau_T}}. \quad (7)$$

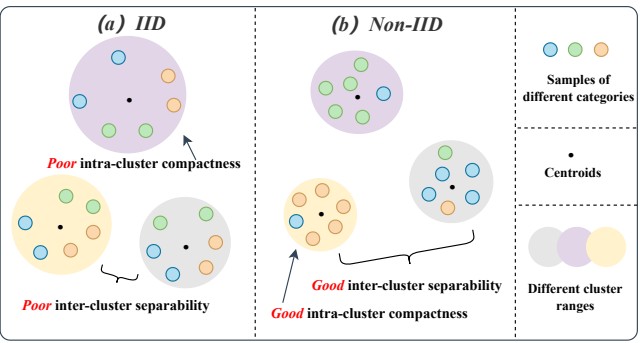

**Figure 3: Motivation of the heterogeneity-aware module. We observed that after pre-training, the concatenated features exhibit relatively poor (good) intra-cluster compactness and inter-cluster separability in the IID (Non-IID) scenarios.**

Therefore, the overall model-contrastive loss is the sum of individual contrastive losses across all clients:

$$\mathcal{L}_T^{(iid)} = \sum_{c=1}^{C} \mathcal{L}_T^{(c,iid)}, \quad \mathcal{L}_T^{(noniid)} = \sum_{c=1}^{C} \mathcal{L}_T^{(c,noniid)}. \tag{8}$$

### 3.4 Heterogeneity-Aware Module

To precisely quantify the boundaries between IID and Non-IID for adaptation to Eq. (8), we design a heterogeneous-aware module. Indeed, the design motivation for this module stems from the following key observation (as shown in Figure 3):

*Key Observation*: During the pre-training phase, as heterogeneity increases, the feature representations obtained by clustering local data on the client side become more similar, resulting in better separability between clusters and compactness within clusters for the global feature concatenated from all features' sets.

Based on this, at the end of the pre-training phase, we apply k-means [34] to the concatenated high-level features. Then we compute the within-cluster sum of squares (*WCSS*) and the average inter-cluster distance (*AICD*) to assess the inter-cluster separability and intra-cluster compactness of the clustering structure:

$$WCSS = \sum_{k=1}^{K} \sum_{j=1}^{n_k} \|\mathbf{h}_{kj} - \mathbf{u}_k\|^2, \tag{9}$$

$$AICD = \frac{1}{K(K-1)} \sum_{i=1}^{K} \sum_{j \neq i}^{K} \|\mathbf{u}_i - \mathbf{u}_j\|, \tag{10}$$

where $\mathbf{u}_k$ denotes the $k$-th centroid of the global clustering, $\mathbf{h}_{kj}$ is the $j$-th concatenated high-level feature belonging to cluster $\mathbf{u}_k$, and $n_k$ is the number of samples in cluster $\mathbf{u}_k$. By considering both *WCSS* (where smaller values indicate better compactness within clusters) and *AICD* (where larger values indicate better separability between clusters), and introducing a threshold $\lambda$, we can calculate the heterogeneity assessment coefficient $A$ for $\mathcal{L}_T$.

$$A = WCSS \,/\, AICD - \lambda. \tag{11}$$

---

**Algorithm 1** Pipeline of HFMVC

**Input:** Dataset with $N$ samples and $M$ views distributed among $C$ clients, with an expectation to be partitioned into $K$ clusters. Global epoch $T$, Local epoch $E$.

**Output:** Global clustering predictions.

1: For each $c \in C$, pretrain its autoencoder.                     ▷ Clients
2: Evaluate the heterogeneity of local data by Eqs. (9) - (12).
3: **while** not reaching $T$ epochs **do**
4:     **for** $c = 1$ to $C$ **do in parallel**
5:         Obtain high-level features and the global model.
6:         **while** not reach the maximum iterations $E$ **do**
7:             Optimize the total loss function by Eq. (13).
8:         **end while**
9:     **end for**
10:     Aggregate models among peer clients by Eq. (2).   ▷ Server
11:     Exchange high-level features among friend clients.
12:     Distribute aggregated models selectively to each client.
13: **end while**
14: Calculate the clustering predictions by Eqs. (14)-(15).

---

Here, we use $\mathbb{I}(\cdot)$ to represent the *indicator function*, and the complete $\mathcal{L}_T$ is expressed as:

$$\mathcal{L}_T = \mathbb{I}(A > 0) \cdot \mathcal{L}_T^{(iid)} + \mathbb{I}(A < 0) \cdot \mathcal{L}_T^{(noniid)}. \tag{12}$$

Eq. (12) indicates the structure and value of the model-contrastive loss are correlated with the level of heterogeneity in the environment, where the quantification of heterogeneity is implemented by the heterogeneity-aware module.

### 3.5 Objective Function

In summary, our objective function can be summarized as:

$$\mathcal{L} = \mathcal{L}_R + \alpha \mathcal{L}_C + \beta \mathcal{L}_T, \tag{13}$$

where $\mathcal{L}_R$, $\mathcal{L}_C$, and $\mathcal{L}_T$ represent the local reconstruction loss of all the clients, the contrastive loss between friend clients, and the model-contrastive loss arising from selective aggregation among peer clients, respectively. Meanwhile, $\alpha$ and $\beta$ denote the corresponding trade-off coefficients. Finally, the server performs k-means on all the high-level features to obtain the global centroids $\mathbf{U}$:

$$\min_{\mathbf{u}_1, \mathbf{u}_2, \ldots, \mathbf{u}_K} \sum_{i=1}^{N} \sum_{j=1}^{K} \|\mathbf{h}_i - \mathbf{u}_j\|^2. \tag{14}$$

Therefore, the final prediction result for sample $i$ is:

$$y_i = \arg\min_j \|\mathbf{h}_i - \mathbf{u}_j\|^2. \tag{15}$$

### 3.6 Pipeline of HFMVC

Algorithm 1 outlines the execution flow for both clients and the server in HFMVC. Once the pretraining is completed, the server assesses the degree of heterogeneity to selectively apply model contrastive learning strategies. Then each client optimizes its local loss. In turn, the server selectively aggregates data from peer clients and shares high-level features with friend clients. These steps form a complete iteration that is repeated until convergence is achieved.

**Table 1: Clustering performance. The mean values (%) of 5 runs are reported. The best and the second best values are highlighted in red and blue.**

| Data | Heterogeneity | Dirichlet (0.5) | | | Dirichlet (1.0) | | | Dirichlet (10) | | | IID, Dirichlet (∞) | | |
|---|---|---|---|---|---|---|---|---|---|---|---|---|---|
| | Metrics | ACC | NMI | ARI | ACC | NMI | ARI | ACC | NMI | ARI | ACC | NMI | ARI |
| MNIST-USPS | DEMVC [43] | 38.72 | 32.38 | 20.34 | 36.42 | 13.28 | 7.35 | 25.97 | 16.68 | 8.07 | 29.39 | 6.31 | 4.06 |
| | SDMVC [44] | 36.57 | 12.96 | 7.02 | 30.01 | 20.79 | 10.96 | 36.93 | 15.53 | 10.10 | 25.16 | 18.26 | 9.16 |
| | MFLVC [45] | 39.74 | 39.55 | 23.11 | 31.89 | 29.65 | 14.88 | 31.88 | 29.65 | 15.25 | 33.36 | 34.71 | 17.44 |
| | GCFAgg [46] | 49.38 | 43.27 | 28.37 | 42.51 | 36.91 | 21.28 | 30.36 | 24.11 | 11.51 | 26.40 | 22.81 | 10.01 |
| | FedDMVC [8] | 46.91 | 41.01 | 28.68 | 43.34 | 31.67 | 25.60 | 29.48 | 21.11 | 11.94 | 26.80 | 20.31 | 10.97 |
| | FCUIF [38] | 45.14 | 38.80 | 24.82 | 31.52 | 23.69 | 12.89 | 25.68 | 17.58 | 8.77 | 23.77 | 16.03 | 7.81 |
| | Ours | 86.21 | 84.66 | 78.96 | 95.32 | 91.07 | 90.29 | 97.35 | 94.04 | 94.31 | 98.27 | 96.04 | 96.31 |
| BDGP | DEMVC [43] | 56.78 | 35.01 | 28.66 | 50.94 | 27.17 | 20.24 | 33.44 | 7.28 | 5.10 | 31.21 | 6.98 | 4.95 |
| | SDMVC [44] | 57.24 | 36.70 | 27.12 | 51.19 | 31.24 | 21.79 | 35.70 | 15.98 | 10.66 | 43.42 | 18.43 | 13.71 |
| | MFLVC [45] | 39.23 | 17.84 | 13.59 | 37.19 | 13.20 | 9.13 | 43.42 | 21.34 | 15.14 | 39.26 | 17.76 | 11.82 |
| | GCFAgg [46] | 50.32 | 26.95 | 21.23 | 46.63 | 23.12 | 18.43 | 33.75 | 9.82 | 6.76 | 34.30 | 11.36 | 7.26 |
| | FedDMVC [8] | 52.92 | 36.56 | 27.51 | 45.48 | 23.17 | 14.13 | 34.40 | 14.08 | 9.05 | 42.64 | 18.79 | 13.23 |
| | FCUIF [38] | 59.76 | 38.89 | 31.36 | 48.01 | 25.18 | 18.31 | 36.59 | 14.44 | 9.38 | 39.62 | 16.58 | 12.46 |
| | Ours | 73.32 | 55.24 | 48.98 | 85.69 | 71.81 | 69.97 | 98.42 | 94.89 | 96.13 | 98.67 | 95.47 | 96.72 |
| Multi-Fashion | DEMVC [43] | 39.17 | 34.50 | 21.13 | 35.39 | 31.98 | 17.11 | 26.58 | 18.85 | 8.97 | 30.45 | 26.77 | 15.12 |
| | SDMVC [44] | 40.16 | 32.86 | 20.53 | 39.45 | 37.83 | 21.58 | 28.86 | 23.17 | 11.53 | 30.98 | 31.55 | 15.87 |
| | MFLVC [45] | 34.00 | 27.68 | 15.65 | 31.72 | 23.49 | 12.56 | 26.51 | 19.87 | 9.43 | 28.32 | 22.46 | 11.34 |
| | GCFAgg [46] | 54.42 | 54.05 | 36.22 | 50.13 | 51.85 | 33.76 | 29.40 | 31.74 | 13.65 | 34.66 | 45.13 | 24.07 |
| | FedDMVC [8] | 33.53 | 28.03 | 15.52 | 34.92 | 28.14 | 16.11 | 29.02 | 23.24 | 11.85 | 38.12 | 38.87 | 23.09 |
| | FCUIF [38] | 46.88 | 25.20 | 19.48 | 48.54 | 25.32 | 18.71 | 36.74 | 13.81 | 9.91 | 40.45 | 17.60 | 12.82 |
| | Ours | 75.08 | 80.30 | 69.08 | 84.41 | 85.28 | 77.18 | 91.27 | 88.61 | 84.44 | 93.18 | 89.80 | 86.85 |
| Caltech-5V | DEMVC [43] | 35.89 | 24.91 | 14.88 | 37.87 | 25.13 | 16.02 | 35.01 | 22.27 | 13.80 | 33.13 | 21.17 | 12.08 |
| | SDMVC [44] | 38.83 | 25.06 | 15.55 | 40.71 | 28.08 | 19.33 | 34.27 | 18.16 | 10.72 | 29.90 | 16.48 | 9.11 |
| | MFLVC [45] | 38.89 | 23.60 | 15.61 | 33.13 | 16.58 | 9.96 | 29.09 | 13.48 | 7.45 | 27.37 | 13.50 | 7.25 |
| | GCFAgg [46] | 42.65 | 30.56 | 18.62 | 41.28 | 30.39 | 17.94 | 31.61 | 22.00 | 11.84 | 34.00 | 24.95 | 14.09 |
| | FedDMVC [8] | 39.93 | 25.56 | 17.66 | 40.39 | 25.39 | 17.50 | 36.27 | 19.13 | 12.25 | 35.30 | 21.67 | 13.29 |
| | FCUIF [38] | 50.83 | 40.67 | 28.53 | 38.67 | 25.97 | 16.38 | 32.09 | 19.15 | 10.75 | 32.56 | 19.30 | 11.70 |
| | Ours | 60.24 | 50.02 | 40.62 | 63.26 | 55.04 | 45.42 | 69.69 | 62.02 | 52.97 | 66.76 | 56.97 | 47.98 |

**Table 2: Ablation experiments of the proposed HFMVC with different heterogeneity settings on BDGP dataset.**

| $\mathcal{L}_R$ | $\mathcal{L}_C$ | $\mathcal{L}_T$ | ACC | NMI | ARI | ACC | NMI | ARI | ACC | NMI | ARI | ACC | NMI | ARI |
|---|---|---|---|---|---|---|---|---|---|---|---|---|---|---|
| | | | Dirichlet (0.5) | | | Dirichlet (1.0) | | | Dirichlet (10) | | | IID, Dirichlet (∞) | | |
| ✓ | | | 50.12 | 28.47 | 16.43 | 41.16 | 27.18 | 14.48 | 68.20 | 52.73 | 31.16 | 73.40 | 55.98 | 41.69 |
| ✓ | ✓ | | 66.04 | 56.25 | 47.35 | 78.28 | 69.89 | 65.60 | 78.80 | 71.59 | 67.04 | 98.24 | 94.50 | 95.68 |
| ✓ | | ✓ | 60.28 | 52.58 | 42.68 | 83.52 | 70.20 | 65.70 | 91.80 | 83.52 | 80.82 | 96.96 | 90.23 | 92.59 |
| | ✓ | ✓ | 67.08 | 53.79 | 46.89 | 76.92 | 70.04 | 65.67 | 81.56 | 78.40 | 72.60 | 81.00 | 79.88 | 73.63 |
| ✓ | ✓ | ✓ | **73.32** | **55.24** | **48.98** | **85.69** | **71.81** | **69.97** | **98.42** | **94.89** | **96.13** | **98.67** | **95.47** | **96.72** |

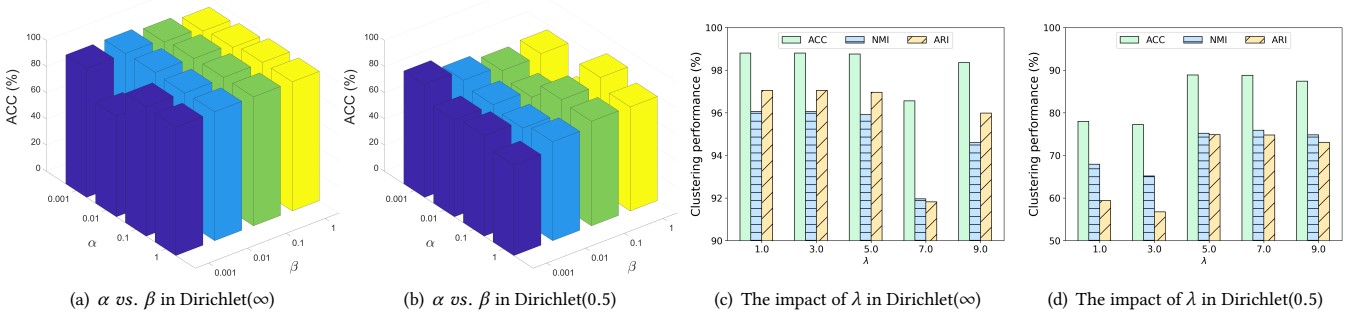

(a) $\alpha$ vs. $\beta$ in Dirichlet($\infty$)          (b) $\alpha$ vs. $\beta$ in Dirichlet($0.5$)          (c) The impact of $\lambda$ in Dirichlet($\infty$)          (d) The impact of $\lambda$ in Dirichlet($0.5$)

**Figure 4: Parameter sensitivity analysis under Dirichlet($\infty$) (IID) and Dirichlet($0.5$) (Non-IID) settings on BDGP.**

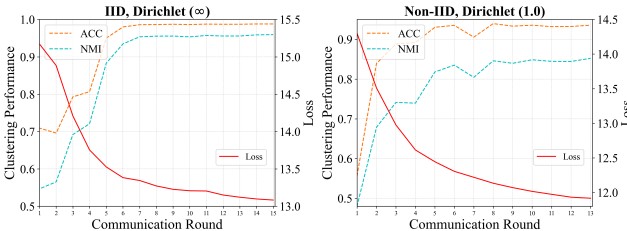

**Figure 5: The convergence curves of HFMVC on BDGP.**

## 4 Experiment

### 4.1 Experimental Settings

*Datasets.* We conduct experiments on four datasets: **MNIST-USPS** [37] (5000 samples with 2 views), **BDGP** [4] (2500 samples with 2 views), **Multi-Fashion** [42] (10000 samples with 3 views) and **Caltech-5V** [13] (1400 samples with 5 views). To better simulate heterogeneous scenarios in the real world, data partitioning adopts the Dirichlet distribution [20]. We configure four heterogeneous environments: Dirichlet (0.5), Dirichlet (1.0), Dirichlet (10), and IID. Here, smaller values within Dirichlet (·) denote greater heterogeneity (imbalances in classes and labels), which is used to evaluate the adaptability of HFMVC and comparative methods.

*Comparison Methods.* We select the following six state-of-the-art (SOTA) methods for comparison: DEMVC [43], SDMVC [44], MFLVC [45], GCFAgg [46], FedDMVC [8] and FCUIF [38]. It is worth noting that among the aforementioned methods, only FedDMVC [8] and FCUIF [38] are applied in a federated environment, while the others are centralized methods. To ensure a fair comparison to the greatest extent, we make simple modifications to the alignment to support a distributed environment. Given the scarcity of research on FedMVC, these modifications are inevitable. Even for FedDMVC and FCUIF, their targeted FL scenarios are limited. In contrast, HFMVC supports distributions with an unlimited number of clients/views. We also set the client-to-view ratio to 5, which means that in a dataset with 2 views, we set up 10 clients accordingly.

*Evaluation Metrics.* We evaluate the clustering performance using three metrics: accuracy (ACC), normalized mutual information (NMI), and adjusted rand index (ARI).

## 4.2 Comparison Results

We conduct experiments of HFMVC alongside other comparative methods on four publicly available datasets, with results presented in Table 1. By comparing the results under four different heterogeneous settings, we can make the following observations:

(1) Our method (HFMVC) consistently demonstrates superior performance and a significant lead across various experimental environments. For example, on the **MNIST-USPS** dataset, HFMVC outperforms in terms of ACC, NMI, and ARI, achieving maximum leads of up to 64.91%, 64.39% and 79.06%, respectively.

(2) Compared to traditional centralized algorithms, HFMVC achieves significant advantages. Furthermore, relative to distributed algorithms like FedDMVC and FCUIF, HFMVC stands out due to its compatibility with multiple clients, stronger cross-client knowledge sharing, and information mining capabilities. Particularly in IID scenarios, the clustering performance of FedDMVC and FCUIF nearly collapses (with ACC on **MNIST-USPS** being respectively only 26.80% and 23.77%, while HFMVC achieves 98.27%), demonstrating the practical feasibility of only HFMVC.

(3) HFMVC exhibits superior robustness. For instance, when the degree of heterogeneity ranges from Dirichlet(0.5) to IID, HFMVC's ACC on **MNIST-USPS** ranges from 86.21% to 98.27%, while on **Caltech-5V**, it ranges from 60.24% to 66.76%. In contrast, other methods exhibit much more significant fluctuations. It is particularly noteworthy that while HFMVC's overall performance tends to improve as the level of heterogeneity decreases, the situation is the opposite for the comparative methods. This seemingly paradoxical phenomenon can be attributed to the following: unlike centralized experimental environments, in distributed settings, data is partitioned into more client nodes, resulting in fewer data points being reconstructed by each autoencoder. As the level of heterogeneity increases, the data reconstructed by single autoencoders becomes purer (with fewer categories), thus yielding better results. HFMVC enhances its clustering performance in IID scenarios through selective aggregation and dual contrastive learning.

### 4.3 Ablation Study

We conduct comprehensive ablation studies on **BDGP** to evaluate the impact of each component, including $\mathcal{L}_R$ (defined in Eq. (1)), $\mathcal{L}_C$ (defined in Eq. (5)), and $\mathcal{L}_T$ (defined in Eq. (12)), with results presented in Table 2. And we can derive the following observations:

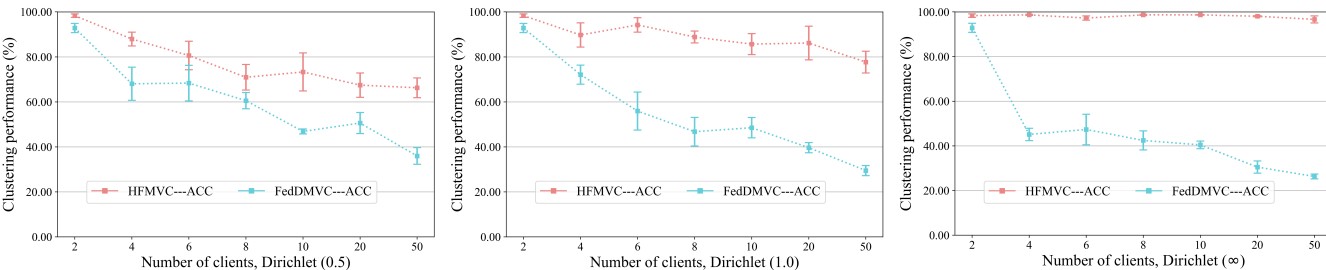

**Figure 6: Clustering performance of HFMVC and FedDMVC under IID and Non-IID scenarios with different numbers of clients.**

(1) An intuitive result is that the method incorporating all components exhibits the best performance, while the result produced solely using $\mathcal{L}_R$ is the poorest, as it only focuses on data reconstruction. This holds true across all levels of heterogeneity.

(2) The configuration lacking $\mathcal{L}_C$ performs worse compared to the setting where all components are used. This discrepancy becomes more pronounced, especially in settings with higher levels of heterogeneity. For example, the difference in ACC between these two settings is 13.04% under Dirichlet(0.5), while it reduces to 1.71% under Dirichlet($\infty$). This is because contrastive learning among clients aligns the representations of the same sample across different views. In other words, the absence of $\mathcal{L}_C$ contributes to poorer clustering results due to the lack of information exchange among **friend clients** (defined in Section 3.1).

(3) The absence of $\mathcal{L}_T$ prevents the global model obtained through selective aggregation from guiding the training of local models on clients. This leads to two issues: firstly, when the degree of heterogeneity is low, the local training process fails to converge quickly, and **peer clients** (defined in Section 3.1) struggle to achieve maximum knowledge sharing. Secondly, when the degree of heterogeneity is high, due to imbalanced samples among clients, there exists significant diversity in the representational capabilities of each autoencoder. However, lacking $\mathcal{L}_T$ often results in a weakened representational capability of local data for each client.

## 4.4 Parameter Analysis

We conduct a detailed analysis of the hyperparameters in HFMVC, including $\alpha$, $\beta$ (defined in Eq. (13)) and $\lambda$ (defined in Eq. (11)). Specifically, this comprises two aspects: **(1)** Figures 4 (a)-(b) illustrate the impact of $\alpha$ and $\beta$ on clustering performance in IID and Non-IID scenarios. It can be observed that the clustering results exhibit overall high ACC values in IID scenarios, indicating good robustness (except when $\alpha$=0.01 and $\beta$=0.001). In Non-IID scenarios, the best performance is observed when both $\alpha$ and $\beta$ are within the range of 0.1 to 1. **(2)** Figures 4 (c)-(d) illustrate the impact of $\lambda$ on clustering performance in IID and Non-IID scenarios. In fact, as a threshold coefficient, $\lambda$ controls the boundary of heterogeneity in the system. Therefore, a smaller/larger value of $\lambda$ may lead to inaccurate perceptions of heterogeneity, thereby affecting clustering performance. It can be observed that when $\lambda$ is 5, all metrics achieve the best performance, regardless of whether it is in IID or Non-IID scenarios. It is worth noting that the value of $\lambda$ varies depending on the dataset.

## 4.5 Convergence Analysis

Figure 5 illustrates the convergence curves of HFMVC under different levels of heterogeneity. It can be observed that under both conditions, HFMVC achieves rapid convergence and eventually stabilizes, indicating its excellent convergence properties. More importantly, the characteristic of rapid convergence significantly reduces the communication overhead of HFMVC.

## 4.6 Scalability Analysis

To measure the scalability of HFMVC, Figure 6 shows the clustering performance of HFMVC as the number of clients changes in environments with varying degrees of heterogeneity. We choose FedDMVC as the comparison method. It can be observed that: **(1)** HFMVC outperforms FedDMVC under all heterogeneity settings; **(2)** In IID scenarios, HFMVC exhibits excellent stability, as increasing the number of clients has little impact on its performance, whereas FedDMVC's performance significantly decreases; **(3)** In Non-IID scenarios, HFMVC's performance decreases as the number of clients increases, but it remains generally more stable than FedDMVC. These observations demonstrate that HFMVC possesses good scalability, mainly because HFMVC promotes the flow of knowledge among clients through selective aggregation and dual contrastive learning, preventing the clustering results from drastically declining with the increase in the number of clients.

## 5 Conclusion

In this paper, we propose a Heterogeneity-aware Federated deep Multi-View Clustering (HFMVC) method, which enables the sharing of multi-view knowledge across clients while preserving privacy. Each client uses autoencoders to perform representation learning, followed by dual contrastive learning between the local and global models as well as among clients to explore consistent and complementary information. Besides, we design a heterogeneity-aware module that adaptively handles different heterogeneous scenarios. Extensive experiments validate that HFMVC exhibits the best performance in both IID and Non-IID environments.

## Acknowledgments

This work is supported by the National Key Research and Development Program of China (2022YFB3105405, 2021YFC3300502).

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
