# OpenReview forum: "Heterogeneity-Aware Federated Deep Multi-View Clustering towards Diverse Feature Representations"
_acmmm.org/ACMMM/2024/Conference — MM2024 Poster_

### Official Review · Reviewer_ozke · 2024-05-22

**Rating:** 4
**Confidence:** 3

**Summary:**

The paper addresses the challenge of clustering analysis in multi-view data, particularly in the context of federated learning where data is distributed across multiple clients. Traditional Multi-View Clustering (MVC) methods assume centralized data storage, which is impractical and raises privacy concerns in real-world scenarios. Federated Learning (FL) offers a solution by aggregating models from different clients without sharing raw data, but integrating FL with MVC to form Federated Multi-View Clustering (FedMVC) introduces new challenges, including data heterogeneity, misalignment of feature representations, and efficient multi-view information mining.

To tackle these issues, the authors propose HFMVC, a Heterogeneity-aware Federated deep Multi-View Clustering method. HFMVC is designed for scenarios where each client holds data from a single view. It employs autoencoders to encode and reconstruct local data, mapping encoded features into a unified high-level space to eliminate irrelevant information. The method addresses the challenges through:
    1.Selective Aggregation :Aggregates local models from clients with the same view data to align feature representations and improve data representation.
    2.Heterogeneity Mitigation: Uses within-cluster sum of squares (WCSS) and average inter-cluster distance (AICD) to measure heterogeneity and guide model-level contrastive learning.
    3.Feature Learning Across Clients: Facilitates the exchange of high-level features via a trusted server and employs dual contrastive learning to enhance multi-view feature learning.
The proposed HFMVC demonstrates robust generalization capabilities, adaptability to both IID and Non-IID scenarios, and state-of-the-art performance in extensive experiments, all while ensuring data privacy. The contributions of this work include the novel federated MVC method, the introduction of a heterogeneity-aware module, and comprehensive experimental validation.

**Strengths:**

1.Integrated Understanding of Information: By perceiving the same object from multiple perspectives, it helps to achieve a more comprehensive understanding of the information. The Multi-View Clustering (MVC) method can explore both consistent and complementary information in the data by promoting collaboration between multiple perspectives, thus achieving a comprehensive clustering structure.

2.Handling Data Heterogeneity: The HFMVC method can assess data heterogeneity and adaptively adjust clustering strategies based on local clustering results. This makes it highly adaptable to both Independent and Identically Distributed (IID) and Non-Independent and Identically Distributed (Non-IID) scenarios.

3.Efficient Feature Learning: By using a trusted server to facilitate the exchange of high-level features and employing a dual contrastive learning module, HFMVC can efficiently perform multi-view feature learning across multiple clients, ultimately uncovering a clearer clustering structure.

4.Experimental Validation: HFMVC has undergone extensive experiments with varying degrees of heterogeneity and different numbers of clients. The results show that HFMVC consistently exhibits state-of-the-art performance while maintaining privacy, demonstrating its effectiveness and robustness in distributed environments.

**Limitations:**

1.Quantify the specific parameters of the transmitted data.
2.How multimodal datasets are specifically divided across different clients.

**Suitability:**

3

---

### Official Review · Reviewer_S1ZD · 2024-05-24

**Rating:** 4
**Confidence:** 3

**Summary:**

In this paper, the authors propose a heterogeneity-aware federated deep multi-view clustering method. It adaptively perceives the degree of heterogeneity and employs contrastive learning to explore consistency and complementary information across clients' multi-view data. Additionally, it seeks consensus among clients whose local data originates from the same view. Furthermore, the authors explain the sample representation logic for local clustering in various heterogeneous environments, propose two metrics to determine the degree of heterogeneity.

**Strengths:**

1. The experimental results in the paper are extensive and show significant improvement over existing methods, demonstrating the superiority of the approach and making a substantial contribution to federated multi-view learning.

2. The method's illustrations are clear, allowing for an easy understanding of the method's structure.

3. The method introduces a noteworthy phenomenon and proposes two metrics based on this phenomenon, contributing to the analysis of IID and non-IID characteristics.

**Limitations:**

1. There seem to be some issues with the legend. The definitions of peer clients and friend clients in Figure 2 appear to differ from those in the text, making it difficult for me to understand their meanings. Additionally, in Representation Learning, are all these features vectors with values 0 and 1? I think these vectors may not be suitable for autoencoders.

2. In my understanding, the optimization of clients is performed simultaneously according to Equation 13. However, in the equation, \(L_C\) seems to require the participation of friend clients, which suggests that friend clients are also being optimized. How is their parallelism ensured? Furthermore, conducting contrastive learning on clients requires substantial computational resources. Considering that clients are generally mobile devices, is it necessary to take into account the computational consumption on the clients?

3. The performance gap between the proposed method and other methods is too large. Could you further explain the specific reasons for this?

4. In the ablation study, there is a variant that removes (L_R). I believe removing this loss function might render the autoencoder ineffective. Are there any other modifications in this variant besides removing (L_R)?

**Suitability:**

3

---

### Official Review · Reviewer_yjJN · 2024-05-25

**Rating:** 4
**Confidence:** 3

**Summary:**

This paper generalizes multi-view clustering to federated learning settings, focusing on addressing heterogeneity and unevenness among data. Although not many new methods have been proposed in terms of multi-view clustering, it has solved some problems in the scenario of federated multi-view clustering. The authors divide clients into peer clients and friend clients. Selective aggregation is applied to peer clients while contrastive learning is conducted for friend clients. Experiments are ample and analysis is accurate, and the method proposed in this paper is far ahead of the comparative methods.

**Strengths:**

1.This paper is detailed and requires a large amount of work
2.The authors acknowledge the challenges present in current federated multi-view clustering algorithms, such as inconsistent feature representation, difficulty in integrating multi-view information, and inability to adapt to heterogeneous data environments.
3.HFMVC adopts strategies like contrastive learning and selective aggregation, enabling adaptive perception of heterogeneity and corresponding adjustments in clustering strategies.

**Limitations:**

1.The innovation of the article is insufficient, and the methods used were proposed in the previous work. Having only one Heterogeneity Aware Module seems insufficient to reflect novelty.
2.The use of WCSS and AICD in the Heterogeneity Aware Module to quantify the boundary between IID and non IID is a bit abrupt. Why did not other clustering indicators be considered, or did mathematical proof or experimental analysis be provided.
3.The baseline of the article experiment itself is not applicable to the federated learning paradigm, and modifications to these methods have not been provided. This comparison may be unfair, which raises doubts about the experimental results. Can author consider integrating multi view methods and federated learning methods as a baseline.
4.In the experimental part, the adopted MNISTUSPS and BDGP have only 2 views, which seems a bit fewer. Moreover, the majority of experiments are based on these two data sets, the results may not be as persuasive as those from datasets with multiple views. Among the six compared methods, only two are in the federated learning environment, it is recommended to increase this number.

**Suitability:**

3

---

### Official Review · Reviewer_j1CA · 2024-05-26

**Rating:** 4
**Confidence:** 3

**Summary:**

The paper propose a novel federated deep multi-view clustering framework to protect local data privacy. The federated setting of this method has wide application scope, Moreover, a heterogeneity-aware module is proposed to the degree of data heterogeneity, while suit iid and non-iid data.  The paper has done extensive experiments to demonstrate its performance.

**Strengths:**

1、	The motivations of the methods are clear and it is easy to understand through the Figure. It is solved to a certain extent through the novel clustering framework.
2、	The experiments section is sufficient and can verify the contribution.
3、	This federated setting that each client owns some sample of a single view has wide application.

**Limitations:**

1	I think WCSS and AICD help to get larger distances between clusters and tighter distances within cluster, but how does it work as a Heterogeneity-Aware Module? Besides, the construction of Eq.12 is unclear.
2	Is the federated setting of your chosen compared federated multi-view methods same to your method?
3	Could you please further explain the difference between Eq.6 and Eq.7 and the reason why the positive pairs and negative pairs of iid and non-iid are different?

**Suitability:**

3

---

### Meta-Review · Area_Chair_7wEX · 2024-06-30

**Recommendation:** Accept (Poster)
**Confidence:** 3

**Metareview:**

The paper presents a novel federated multi-view clustering framework, HFMVC, which addresses key challenges in multi-view clustering algorithms. The proposed method effectively handles inconsistent feature representation, integrates multi-view information, and adapts to heterogeneous data environments. The experimental results are extensive, demonstrating significant improvements over existing methods and illustrate the method's robustness and applicability.
Despite some areas that could benefit from further clarification and expansion, the reviewers appreciated the authors' rebuttal and argued for acceptance of the paper.